# Outlook for 615 Small Intestinal Neuroendocrine Tumor Patients: Recurrence Risk after Surgery and Disease-Specific Survival in Advanced Disease

**DOI:** 10.3390/cancers16010204

**Published:** 2024-01-01

**Authors:** Cecilie Slott, Seppo W. Langer, Stine Møller, Jesper Krogh, Marianne Klose, Carsten Palnæs Hansen, Andreas Kjaer, Pernille Holmager, Rajendra Singh Garbyal, Ulrich Knigge, Mikkel Andreassen

**Affiliations:** 1ENETS Center of Excellence, Copenhagen University Hospital—Rigshospitalet, 2100 Copenhagen, Denmark; cecilie.slott@regionh.dk (C.S.); seppo.langer@regionh.dk (S.W.L.); stine.moeller.02@regionh.dk (S.M.); jesper.krogh@dadlnet.dk (J.K.); marianne.christina.klose.01@regionh.dk (M.K.); carsten.palnaes.hansen@regionh.dk (C.P.H.); akjaer@sund.ku.dk (A.K.); pernille.holmager.01@regionh.dk (P.H.); rajendra.singh.garbyal@regionh.dk (R.S.G.); ulrich.peter.knigge@regionh.dk (U.K.); 2Department of Endocrinology and Metabolism, Copenhagen University Hospital—Rigshospitalet, 2100 Copenhagen, Denmark; 3Department of Clinical Medicine, University of Copenhagen, 2200 Copenhagen, Denmark; 4Department of Oncology, Copenhagen University Hospital—Rigshospitalet, 2100 Copenhagen, Denmark; 5Department of Surgery and Transplantation, Copenhagen University Hospital—Rigshospitalet, 2100 Copenhagen, Denmark; 6Department of Clinical Physiology and Nuclear Medicine & Cluster for Molecular Imaging, Copenhagen University Hospital—Rigshospitalet, 2100 Copenhagen, Denmark; 7Department of Biomedical Sciences, University of Copenhagen, 2200 Copenhagen, Denmark; 8Department of Pathology, Copenhagen University Hospital—Rigshospitalet, 2100 Copenhagen, Denmark

**Keywords:** small intestinal neuroendocrine tumor, survival, prognosis, recurrence

## Abstract

**Simple Summary:**

This large single-center study of 615 patients with small intestinal neuroendocrine tumors (siNET) showed that the risk of recurrence after intended radical surgery was high with a recurrence-free survival rate of 40% after 10 years follow-up, and that late recurrencies were frequent. For cases with disseminated disease, the median disease specific survival was around 7 years. Tumor proliferation expressed by Ki-67 index was identified as a prognostic factor whereas proliferation expressed by WHO grade did not predict prognosis. In conclusion, radical intended surgery recurrence rates were high thereby justifying long-term follow-up. Proliferation expressed by Ki-67 index as a continuous variable, rather than grouped according to WHO grading, was an independent prognostic factor for both recurrence-free survival and disease-specific survival supporting the need for reevaluation of the existing grading system.

**Abstract:**

Background: Small intestinal neuroendocrine tumors (siNET) are one of the most common neuroendocrine neoplasms. Radical surgery is the only curative treatment. Method: We utilized a single-center study including consecutive patients diagnosed from 2000 to 2020 and followed them until death or the end of study. Disease-specific survival and recurrence-free survival (RFS) were investigated by Cox regression analyses with the inclusion of prognostic factors. Aims/primary outcomes: We identified three groups: (1) disease specific-survival in the total cohort (group1), (2) RFS and disease-specific survival after intended radical surgery (group2), (3) disease specific-survival in patients with unresectable disease or residual tumor after primary resection (group3). Results: In total, 615 patients, with a mean age (SD) 65 ± 11 years were included. Median (IQR) Ki-67 index was 4 (2–7)%. Median disease-specific survival in group1 was 130 months. Median RFS in group2 was 138 months with 5- and 10-year RFS rates of 72% and 59% with age, plasma chromogranin A (p-CgA) and Ki-67 index as prognostic factors. The ten year disease-specific survival rate in group2 was 86%. The median disease-specific survival in group3 was 85 months with age, Ki-67 index, p-CgA and primary tumor resection as prognostic factors. When proliferation was expressed by WHO grade, no difference was observed between G1 vs. G2 for any of the primary outcomes. Conclusions: Recurrence rates remained high 5–10 years after surgery (group2) supporting long-term follow-up. Median disease-specific survival in patient with unresectable disease (group3) was 7 years, with a favorable impact of primary tumor resection. Our data does not support the current grading system since no significant prognostic information was detected in G1 vs. G2 tumors.

## 1. Introduction

Gastroenteropancreatic neuroendocrine neoplasms (GEP-NEN) represent a heterogeneous group of rare neoplasms arising from neuroendocrine cells. GEP-NEN are classified into well-differentiated neuroendocrine neoplasm, i.e., neuroendocrine tumors (NET), and poorly differentiated neoplasm, i.e., neuroendocrine carcinomas (NEC). NET is further classified based on proliferation index as low-grade (NET G1; Ki-67 < 3%), intermediate-grade (NET G2; Ki-67 3–20%) or high-grade (NET G3; Ki-67 > 20%) [1,2].

The small intestine is one of the most common sites of origin of GEP-NEN [3,4]. Small intestinal neuroendocrine tumors (siNET) produce serotonin and other vasoactive substances. If released from liver metastases, they may lead to carcinoid syndrome [5,6]. The proliferation index is often low with the vast majority of tumors being G1 or G2 [3,7,8]. According to the US Surveillance, Epidemiology, and End Results (SEER) database the age-adjusted incidence of siNET has increased from 0.20 (1973) to 1.25 (2012) per 100,000 inhabitants [4].

Intended radical surgery is the only potentially curative treatment and is recommended for both local and metastatic disease [9,10]. The ratio of patients with disseminated disease at diagnosis varies from 40 to 75% [3,11]. Due to the slow growth of the tumor, recurrence may occur many years after the surgery [12,13]. Risk factors for disease recurrence include disease stage and proliferation index [12,13,14]. If radical surgery is not possible, systemic therapy with somatostatin analogues (SSA) is recommended as the first-line treatment in low-grade tumors. Other treatment options include somatostatin receptor-targeting peptide receptor radionuclide therapy (PRRT), everolimus, as well as local therapy such as embolization of liver metastases or percutaneous ultrasound guided radio- or microwave ablation [9,15,16]. The impact of palliative surgery on prognosis in patients with disseminated disease, including resection of primary tumor, is debated [5,17,18,19,20].

A recently published meta-analysis showed that the overall survival has improved over the last few decades [21]. For metastatic siNET, the 5- and 10-year overall survival rates were 37 and 67%, respectively [21]. However, the data was collected from 1960 to 2018 and may not represent modern treatment algorithms, including the use of PRRT which was introduced in European centers more than 20 years ago and only recently was approved and made available in the US [21,22,23].

The aim of this study was to describe the prognosis of a large cohort of siNET patients treated at the European Neuroendocrine Tumor Society (ENETS) Center of Excellence at Copenhagen University Hospital, Rigshospitalet from 2000 to 2020. Outcomes included: (1) disease-specific survival in the total cohort, (2) Recurrence-Free Survival (RFS) and disease-specific survival after intended radical surgery, and (3) disease-specific survival in patients with unresectable disease or residual tumor after resection.

## 2. Method

The Copenhagen ENETS Center of Excellence covers 2,700,000 inhabitants corresponding to approximately half of the Danish population. The NET center was established in 2009 among the first six centers of excellence in Europe.

### 2.1. Study Population

Consecutive siNET patients referred to Rigshospitalet from 1 January 2000 to 31 December 2020 were included in the study. The diagnosis was based on positive immunostaining for serotonin from primary tumor or metastasis. At the time of diagnosis, all specimens were reevaluated by specialized NET pathologists. All patients had a somatostatin receptor imaging (SRI) performed at diagnosis, initially as an octreotide scintigraphy and a computer tomography (CT), and from 2012 and onwards, with ^64^Cu-DOTATATE or ^68^Ga-DOTATOC, positron emission tomography (PET) and CT.

### 2.2. Data Acquisition

The following data were prospectively collected from the electronic patient file and stored in a database: day of diagnosis, demographic data, immunohistological characteristics (Ki-67 index, chromogranin A (CgA), serotonin and synaptophysin) and biochemistry (plasma CgA (p-CgA), 5-hydroxyindoleacetic acid (5-HIAA) from urine or plasma), resection of the primary tumor (radical surgery or not), and grading. Diagnosis of NEC was based on a Ki67 proliferation index > 20%, histological morphology and if investigated with possible associated p53-mutation [1,24]. Since the establishment of the grading system based on proliferation by Rindi et al. in 2006, we have adhered to this methodology [25,26]. In all analyses, the results of urine- or plasma-5-IHAA were dichotomized with the upper-normal level as the cut-off.

As part of the study protocol, the following additional data were collected from patient files: metastases at diagnosis, surgery of metastases, treatment modalities, concomitant cancer diagnoses, presence of carcinoid heart disease, time to recurrence after radical surgery, time to death, cause of death and time to last follow up. Staging at diagnosis was divided into 4 groups: (1) primary tumor(s) without metastases, (2) primary tumor(s) with regional mesenteric lymph node metastases, (3) intra-abdominal metastases, including liver and other metastases, carcinosis, and retroperitoneal lymph nodes, and (4) extra-abdominal metastases. Due to nationwide electronic access to patient files, no patients were lost to follow-up in the survival analysis. The patients were followed until their death or the end of the study (31 December 2021).

### 2.3. Outcome

Group1: total cohort

The primary outcome was median disease-specific survival. Secondary outcomes were disease-specific survival rates at 2, 5, 10 and 15 years after diagnosis and prognostic factors.

Group2: patients who underwent intended radical surgery with no detectable tumor tissue on SRI performed 1–3 months post-surgery. Primary outcomes were median RFS and median disease-specific survival. Recurrence was defined as the detection of tumor recurrence through imaging and/or confirmed by histological analysis, or alternatively, as instances leading to disease-specific death. Secondary outcomes were RFS and disease-specific survival rates at 2, 5, 10, and 15 years after diagnosis and prognostic factors.

Group3: patients with unresectable disease, patients who refused surgery, patients who underwent palliative surgery or patients who underwent intended radical surgery of the primary tumor and/or metastasis but had residual tumor tissue on SRI performed 1–3 months post-surgery.

The primary outcome was median disease-specific survival. Secondary outcomes were disease-specific survival rates at 2, 5, 10 and 15 years after diagnosis and prognostic factors.

### 2.4. Statistics

Statistical analyses were performed using the statistical program SPSS (IBM, version 28.0.0.0, Chicago, IL, USA). Data are presented as median and interquartile range (IQR) or mean (SD). Plasma CgA and Ki-67 index were logarithmically transformed (Log2) in all analyses. Baseline data are presented for the entire cohort and subsequently divided into the two subgroups. Comparison between the groups was performed using the Chi-square test for categorical variables, and the independent *t*-test for continuous variables.

RFS and disease-specific survival were assessed using Kaplan–Meier curves stratified by WHO grade and stage at diagnosis. Furthermore, disease-specific survival was stratified by primary tumor resection in the subgroup of patients with disseminated disease (group3). Differences were assessed by log-rank test. Cox regression analyses were used to identify potential risk factors for RFS and disease-specific survival, respectively. The following potential risk factors were investigated in univariate analyses; gender, age, year of diagnosis (from 2000 to mid-2010 vs. mid 2010 to 2020), stage at diagnosis, proliferation (expressed both by grade and by Ki-67 index) concomitant cancer diagnoses before and after siNET diagnosis, p-CgA, urine/plasma 5-HIAA, and presence of carcinoid heart disease. Subsequently, all covariates with *p*-value < 0.15 were included in a multivariable Cox regression analysis using backstep elimination (conditional method). A second multivariate Cox regression analysis was carried out with proliferation expressed as Ki-67 index instead of WHO grade. The results of the Cox regression models are expressed as hazard ratios (HR) and 95% confidence interval. HR for p-CgA and Ki-67 index were estimated per 2-fold increase in non-logarithmically transformed p-CgA and Ki-67 index. A *p*-value < 5% was considered significant for all analyses.

## 3. Results

### 3.1. Patient Characteristic at Diagnosis

Table 1 presents the baseline characteristics of the patients. A total of 615 patients were included with a mean age of 65 (±11) years at diagnosis. There was an increase in the number of patients diagnosed in the latter half of the inclusion period (from mid-2010 to 2020) from 190 to 425. This corresponds to an annual change in incidence from eight cases per million to 16 cases per million. The proportion of patients eligible for radical surgery (group2) was 39% in the first period and 35% in the latter (*p* = 0.29). Forty-seven patients (8%) had local disease, 220 (36%) had mesenteric lymph node metastases, 271 (44%) had disseminated intra-abdominal disease, and 77 (12%) had extra-abdominal metastases. A total of 392 (64%) patients underwent primary tumor resection with either a curative intended or to alleviate local symptoms. The median Ki-67 index was 4 (2–7)%. Patients eligible for radical surgery (group2) had a lower Ki-67 index, compared to patients with disseminated disease (group3), (3 (2–5) vs. 5 (2–9)%, *p* < 0.001) and they were younger (64 ± 12 vs. 66 ± 11 years, *p* = 0.003).

### 3.2. Group1 (Total Cohort) Disease-Specific Survival

In total, 192 (31%) patients died from siNET with a median disease-specific survival of 130 months. The median follow-up time was 52 (24–92) months. The 2-, 5-, 10- and 15-year disease-specific survival rates were 89%, 77%, 54%, and 44%, respectively (Figure 1A). Significant risk factors in multiple variable analysis were age at diagnosis (HR = 1.04, 95%CI: 1.03–1.06; *p* < 0.001), intended radical surgery (reference is surgery, HR 3.6; 95%CI: 2.0–6.5; *p* < 0.001), stage at diagnosis (pairwise comparison showed significant differences (*p* < 0.001) only between local disease and extra-abdominal metastasis), and WHO grading (pairwise comparison showed significant (*p* < 0.001) differences only between G1 and NEC. For details see Table 2 and Figure 1B and Figure 2A). When replacing the categorical grade (WHO classification) with the Ki-67 index proliferation remained an independent risk factor (HR 1.2 per 2-fold-increase in Ki-67 index; 95%CI: 1.1–1.4; *p* < 0.001) (Table 2).

The median overall survival for the total cohort was 94 (42–189) months.

### 3.3. Group2—Intended Radical Surgery

#### 3.3.1. Recurrence-Free Survival

Among 224 patients who underwent intended radical surgery, 69 (31%) had recurrence during a median follow-up of 71 (35–120) months. One patient was lost to follow-up due to immigration. The median RFS was 138 months and the 2-, 5-, 10- and 15-year RFS rates were 91%, 72%, 59% and 41%, respectively (Figure 1C). In the multivariable analysis (Table 3) stage at diagnosis (*p* < 0.016, for pairwise comparison see Table 3 and Figure 1D and Figure 2B), p-CgA (HR pr. 2-fold increase in p-CgA 1.3; 95%CI: 1.1–1.5; *p* = 0.001) and Ki-67 index (HR pr. 2-fold increase in Ki-67 1.4; 95%CI: 1.1–1.9; *p* = 0.01) were identified as significant risk factors. Proliferation expressed by WHO grading did not predict recurrence (Table 3). The two latest recurrences occurred at 177 and 169 months of follow-up (Figure 1C).

#### 3.3.2. Disease-Specific Survival

Disease-specific death was observed in 20 (9%) patients. The 2-, 5-, 10- and 15-year disease specific survival rates were 98%, 94%, 86% and 80%, respectively (Figure 1E). In the multivariable analysis, stage at diagnosis (*p* = 0.02, for pairwise comparison see Table 4) and p-CgA (HR 1.3 per 2-fold increased; 95%CI: 1.0–1.7; *p* = 0.05) were identified as independent prognostic factors for disease specific survival. For details see Table 4 and Figure 1F and Figure 2C.

### 3.4. Group3—Unresectable Disease or Residual Tumor after Resection

#### Disease-Specific Survival

Among the 391 patients with disseminated disease, a total of 172 (44%) died due to siNET, during a median follow-up time of 46 (19–76) months. The median disease-specific survival was 85 (45–152) months. The 2-, 5-, 10- and 15-year disease-specific survival rates were 83%, 66%, 35%, and 22%, respectively (Figure 1G and Figure 1H). In the multivariable analysis (Table 5), the following variables were identified as independent risk factors for disease-specific survival; age at diagnosis (HR 1.04; 95%CI: 1.0–1.1; *p* < 0.001), WHO grading (in pairwise comparison with G1 as reference, the only significant (*p* < 0.001) difference was between G1 and NEC (Table 5 and Figure 2D) and p-CgA (HR per 2 fold increase 1.2; 95%CI: 1.2–1.3; *p* < 0.001). When replacing WHO grade with Ki-67 index, age and p-CgA remained significant risk factors along with primary tumor resection (resection is reference HR 1.5; 95%CI: 1.0–2.3; *p* = 0.041, Figure 3) and Ki-67 index (HR 1.3 per 2-fold increase in Ki-67 index; 95%CI: 1.1–1.5; *p* < 0.001).

Patients with unresectable disease received different standard medical treatments as listed in Table 6. There have been minor changes in treatment strategi over time, e.g., increased use of everolimus in NETG2 and reduced use of streptozotocin/5-fluorouracil. However, throughout most of the study period, somatostatin analogues were used as first-line and PRRT as second-line therapy for most patients with NETG1/G2.

## 4. Discussion

This is the largest single-center study to date reporting data on prognosis in siNET patients. The major findings were that one-third of the patients were eligible for intended radical surgery (group2), but they faced a very high risk of late recurrences, with a median RFS of 12 years. In patients with unresectable disease (group3) the median disease-specific survival was 7 years with age, with p-CgA and palliative resection of primary tumor identified as prognostic factors. The data support the use of Ki-67 index as a prognostic factor but raise questions about the current WHO grading system which separates G1 and G2 with a cutoff at 3%, as it provided little prognostic information.

Around two-thirds of the patients presented with disseminated metastatic disease at the time of diagnosis, with 12% having extra-abdominal metastases. Previous publications have reported divergent results with lymph node metastases occurring in 25–76% of patients and distant metastases in approximately 10% of patients [12,13]. Our study confirmed low proliferation in siNET with a median Ki-67 of 4%. One-third of the tumors were classified as NET G1 tumors, while approximately two-thirds were classified as NET G2 tumors. Less than 3% of cases had Ki-67 > 20%. The Ki-67 index observed in our cohort was higher than that reported in most previous publications, including a meta-analysis reporting that 60% of patients had NET G1 [12,27]. Different patient selection criteria among studies complicates the comparison with our cohort, which includes the full spectrum of the disease. However, we cannot exclude the possibility that the tumors in our cohort had higher proliferation which may have some implications for prognosis. The mean age at diagnosis in our cohort was 65 years consistent with results from recently published reviews [12,21].

In the entire cohort, the 5-year and 10-year disease-specific survival rates were 77% and 54%, respectively. Our analyses identified age, intended radical tumor resection, stage, and Ki-67 index as independent prognostic factors for disease-specific survival. However, when proliferation was expressed in grade, we only observed a significant difference between NET G1 and NEC. It is important to note that 96% of our patients had NET G1 or NET G2 tumors and within this sub-group grading did not provide any prognostic prediction (HR 1.05, *p* = 0.80). These findings contrast with a previous meta-analysis that demonstrated a worse prognosis in NET G2 compared to NET G1 tumors [13].

Approximately one-third of the patients who underwent intended radical surgery had recurrence during the follow up period. The median RFS was approximately 12 years and the 5-year RFS rate was 72%. Prognostic factors for recurrence included stage at diagnosis, p-CgA and Ki-67 index while tumor grade did not predict recurrence. The presence of mesenteric lymph node metastases was associated with a threefold increased risk of recurrence compared to patients without lymph node metastases. The presence of other intra-abdominal metastases further increased the HR to 6, and in the few patients with extra-abdominal metastasis who underwent intended radical surgery the risk was further elevated with a HR of 9. The time to recurrence observed in this study was substantially longer than that reported in previous studies, where the median RFS ranged between 2 and 9 years [12,28,29,30]. Differences in criteria for intended radical surgery and surgical approach might contribute to these variations. In our region, surgery for siNET is always considered as first-line treatment and may be performed as open or laparoscopic surgery dependent on the anatomical circumstances. As depicted in the Kaplan–Meier curves (Figure 1C,D and Figure 2B), recurrence occurred across all stages and without any apparent flattening of the curves, with the latest relapse occurring nearly 15 years after initial diagnosis. Disease-specific death after intended radical surgery was only observed in 9% of patients with a disease-specific survival rate of 80% after 15 years of follow-up. Long-term follow-up is recommended by both ENETS and NANETS guidelines [16,31,32,33]. ENETS guidelines suggest cross-sectional imaging every 6–12 months for 10 years with an individualized decision to continue thereafter [32]. Taken together, in our opinion, standard risk factor evaluation using markers such as p-CgA, grade and stage does not allow an individualized follow-up algorithm after intended radical surgery. However, considering the favorable prognosis in this patient group, frequent use of CT- based imaging raises concerns regarding radiation exposure, worries for the patients and resource utilization. Based on our limited data, the benefit of follow-up beyond 15 years can be debated. Additionally, for younger patients, magnetic resonance imaging as a surveillance tool appears to be a justified approach to minimize radiation exposure. Further studies are needed to identify patients at risk of recurrence.

In patients with unresectable disease (group3), the median disease-specific survival was 7 years, with a 5-year survival rate of 66%. Age, primary tumor resection, p-CgA and proliferation both expressed by Ki-67 index and WHO grade (in pairwise comparison only G1 vs. NEC was significant) were identified as prognostic factors. Similar findings have been reported in other recent studies, including a meta-analysis of metastatic siNET which found 5-year survival of 67–72% and identified age and grade as prognostic factors [7,21]. An interesting observation in our study was that primary tumor resection was associated with improved disease-specific survival, both in univariate and multivariate analysis. As depicted in Figure 3, the favorable effect of primary tumor resection was evident throughout the study with 10-year survival rate of 50% in patients who underwent primary tumor resection compared to 24% in patients who did not undergo primary tumor resection. The prognostic impact of primary tumor resection is still a subject of debate. Previous studies including a recently published meta-analysis have reported better prognosis consistent with our result [7,20,34], however this result was not supported by a large well-conducted Swedish study reporting no survival advantage of prophylactic primary tumor resection [20].

All patients were offered medical treatment in agreement with relevant guidelines [9,15,16]. The prognosis of patients with disseminated disease did not improve during the study period (2000–2009 vs. 2010–2020). It is not possible to evaluate the impact of PRRT on prognosis since PRRT were offered throughout most of the study period. Among patients diagnosed between 2000 and 2009, 44 (23%) received PRRT compared to 96 (23%) in the latter period.

Guidelines recommend measurements of p-CgA as a disease marker during follow-up. Our data support that levels of p-CgA provides prognostic information, but its use in everyday clinical practice has been questioned due to its low sensitivity and specificity [35]. Urine- and plasma-5-IHAA did not provide prognostic information for any of the outcomes.

Proliferation expressed by Ki-67 index was identified as an independent factor for worse outcome in all our outcomes. WHO grade was also a significant prognostic factor for disease-specific survival in the total cohort and for disease-specific survival in patients with unresectable disease. However, pairwise comparison did not show a significant difference between NET G1 vs. NET G2 or NET G1 vs. NET G3. Therefore, our data question the value of the current grading system that uses a Ki-67 cut-off of 3% as a prognostic tool. A study from 2023 has displayed significant differences in overall survival and RFS between NET G2 divided into low (3–9%) grade and high (10–20%) grade [14]. Other previous studies have suggested that a cut-off level of 5% provides better prognostic information [8,13]. We are currently planning a separate publication that will specifically focus on the Ki-67 index cut-offs. The forthcoming study will combine data from the current cohort with a large cohort of patients with pancreatic NET. As an important limitation, it should be noted that differences in treatment strategies in NET G1 vs. NET G2 might contribute to improved prognosis in G2. Only a few patients with NET G3 or NEC were included, but those with NEC seemed to have a very poor prognosis (Figure 2A) supporting the new classification that distinguishes between NET G3 and NEC. Future studies of siNET are needed to better classify low- and intermediate-grade tumors for more accurate guidance on surveillance, treatment strategies and prognosis.

In our cohort, the incidence of siNET doubled from 2000 to 2009 to 2010 to 2020, but without a change in prognosis or stage at diagnosis. As one hypothesis, the rise in incidence has been attributed to improved and more frequent use of imaging, leading to incidental discoveries of siNET. However, we have previously shown that the proportion of patients incidentally diagnosed in 2010–2011 vs. 2019–2020 was unchanged in our center [36]. Thus, there may be a genuine increase in siNET incidence together with other contributing factors such as increased awareness among pathologists and clinicians.

The major strength of this study lies in the large cohort and extensive follow-up period. In the metanalysis from 2022 [21], 23 studies were identified including 17 single-center studies, each with a smaller cohort then the one presented here. Moreover, only eight of the 23 studies reported 10-year survival data and among these only the SEER-database had a larger cohort [37]. Another advantage of our study was the prospective collection of most data and the establishment of a well-defined cohort which consisted exclusively of siNET patients. The nationwide file system in Denmark ensured no loss to follow-up in the survival analysis. Furthermore, all specimens were reevaluated at diagnosis by a pathologist specialized in NET. While we did not conduct a pathological re-examination as part of our study protocol, it is noteworthy that since Capella et al.’s original prognostic classification in 1994, the primary changes have only been the introduction of a grading system based on proliferation in 2006, and the subsequent differentiation of high-grade NET G3 from NEC [25,26,38].

As for study limitations, some of the data were retrospectively collected, resulting in incomplete data, particularly from the early years. We cannot exclude the possibility that a few patients might have refused referral or were not candidates for referral due to poor performance, potentially leading to an underestimation of the incidence and disease severity. Unrecognized residual confounding is also a concern. For example, the better prognosis observed after resection of the primary tumor could be attributed to differences in performance status or dissemination that was not detected when interpreting the calculated prognosis after 10- and 15-year follow-up. Finally, the transition from traditional gamma camera-based somatostatin receptor scintigraphy to PET technology with ^68^Ga- DOTATOC/^64^Cu-DOTATATE-PET has likely improved the detection of metastases over time [39].

## 5. Conclusions

In conclusion, patients who underwent surgery relapsed many years after intended radical surgery. The median RFS observed in our study was longer than previously reported, highlighting the significance of carefully selecting appropriate patients for surgery. The limited predictive value of current prognostic factors for recurrence of siNET does not allow individual follow-up regimens and the observed late recurrence supports long-term follow up of this patient groupKi-67 index as a continuous variable, rather than grouped according to WHO grading, was an independent prognostic factor for both RFS and disease-specific survival supporting the need for reevaluation of the existing grading system particular with respect to the cut off between G1 and G2. Multigene blood analysis has shown promising results for the prediction of recurrence but has yet to be integrated into everyday clinical practice. Further studies are warranted to identify additional prognostic factors that can enhance the precision of surveillance and treatment strategy guidance [40,41].

## Figures and Tables

**Figure 1 cancers-16-00204-f001:**
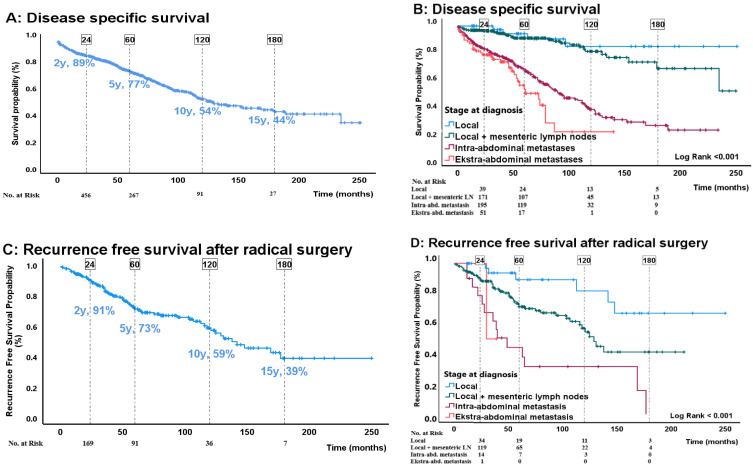
(**A**–**H**) Kaplan–Meier Survival Curves showing 2-, 5-, 10 and 15-year survival/RFS (survival probability (%)) in months, overall and stratified for stage at diagnosis. Differences in stages were assessed by log-rank test.

**Figure 2 cancers-16-00204-f002:**
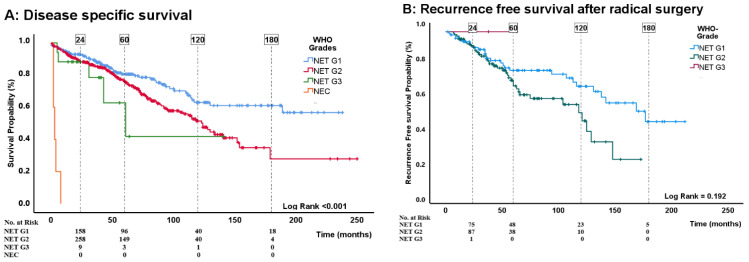
(**A**–**D**) Kaplan–Meier survival curves showing, 2-, 5-, 10- and 15-year survival/RFS (survival probability (%)) in months stratified by WHO grade. Differences in WHO grades were assessed by log-rank test.

**Figure 3 cancers-16-00204-f003:**
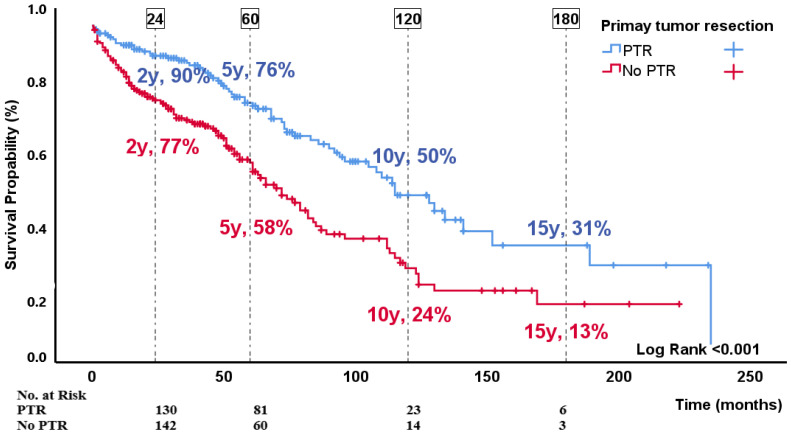
Disease-specific survival (group3—unresectable disease or remnant tumor on first postoperative imaging) stratified for primary tumor resection. Difference was assessed by log-rank test. PTR: primary tumor resection.

**Table 1 cancers-16-00204-t001:** Characteristic. Patient characteristics divided in total cohort group1, group2 (radical surgery) and group3 (patients with unresectable disease or remnant tumor on first postoperative imaging). Biochemical variables are expressed as median (IQR). ^a^ Normal level is 30–130 pmol/L. ^b^ Normal level is < 40 µmol/day. ^c^ Normal level is 35–123 nmol/L. Data on grade was not available in 41 of the 615 patients, primarily due to lack of Ki-67 staining in the early years. Significant *p*-values are marked with bold.

		All Patients, Group1 (n = 615)	Group2 (n = 224)	Group3 (n = 391)	*p*-Value Gr 2 vs. Gr 3
Age, years (mean (SD))		65 (±11)	63 (±12)	66 (±11)	**0.002**
Year of diagnosis					0.293
	2000–mid-2010	190 (31)	75 (39)	115 (61)	
	Mid-2010–2020	425 (69)	149 (35)	276 (65)	
Gender, n (%)					
	Female (%)	288 (47)	106 (47)	182 (47)	0.853
Primary tumor resection, n (%)		392 (64)	224 (100)	168 (43)	**<0.001**
WHO grades NET (n = 574), n (%)					**<0.001**
	NET G1	203 (35)	103 (46)	100 (26)	
	NET G2	349 (61)	111 (49)	238 (61)	
	NET G3	17 (3)	3 (2)	14 (3)	
	NEC	5 (1)	0 (0)	5 (1)	
Stage at diagnosis, n (%)					**<0.001**
	Local	47 (8)	43 (19)	4 (1%)	
	Local + mesenteric lymph nodes	220 (36)	159 (71%)	61 (16%)	
	Intra-abdominal metastases	271 (44)	20 (9%)	251 (64%)	
	Extra-abdominal metastases	77 (12)	2 (1%)	75 (19%)	
Ki-67, %, (median (IQR)) (n = 574)		4 (2–7)	3 (2–5)	5 (2–9)	**<0.001**
Plasma CgA, pmol/L ^a^ (n = 568)		402 (134–1558)	138 (82–317)	909 (272–2838)	**<0.001**
Urin 5-HIAA, umol/day ^b^ (n = 313)		46 (24–196)	24 (17–37)	84 (37–298)	**<0.001**
Plasma 5-HIAA, nmol/L ^c^ (n = 26)		179 (83–419)	92 (81–210)	274 (84–841)	**0.012**

**Table 2 cancers-16-00204-t002:** Prognostic factors for disease-specific survival in the total cohort (Group1). Uni- and multivariable analysis of prognostic factors for disease-specific survival in the total cohort. Only variables with *p*-value in univariate analysis < 0.15 (except age) are included in the table and in the multivariable analyses. Hazard ratio (HR) for plasma chromogranin A (p-CgA) and Ki-67 index were estimated per 2-fold increase in non-logarithmically transformed p-CgA and Ki-67 index. Because both WHO grade and Ki-67 index were significant risk factors in univariable analysis, two different multivariable analyses were conducted. CI: confidence interval.

**Variables**		**Univariable Analysis**		
		** *HR* **	** *95%CI* **	** *p-Value* **
Age at diagnosis (n = 615)		1.04	1.0–1.1	<0.001
Radical surgery (ref: yes) (n = 615)		7.7	4.8–12.2	<0.001
Stage (n = 615)				<0.001
	Local (n = 47)	1.0		
	Local and mesenteric lymph nodes (n = 220)	1.8	0.6–5.1	0.27
	Intra-abdominal metastases (n = 271)	8.1	3.0–22.1	<0.001
	Extra-abdominal metastases (n = 77)	12.1	4.2–34.7	<0.001
WHO Grading (n = 574)				<0.001
	NET G1 (n = 203)	1.0		
	NET G2 (n = 349)	1.6	1.1–2.2	0.001
	NET G3 (n = 17)	2.4	1.0–6.1	0.06
	NEC (n = 5)	54.0	19.7–147.9	<0.001
Ki-67% (n = 572)		1.4	1.3–1.6	<0.001
**Variables**		**Multivariable Analysis (WHO Grade)**		
		** *HR* **	** *95%CI* **	** *p-Value* **
Age at diagnosis		1.04	1.03–1.06	<0.001
Radical surgery (ref: yes)		3.6	2.0–6.5	<0.001
	Stage			<0.001
	Local	1.0		
	Local and mesenteric lymph nodes	1.2	0.4–3.4	0.78
	Intra-abdominal metastases	2.9	1.0–8.5	0.06
	Extra-abdominal metastases	3.7	1.2–11.6	0.025
WHO Grading				<0.001
	NET G1	1.0		
	NET G2	1.0	0.7–1.5	0.79
	NET G3	1.5	0.6–3.9	0.38
	NEC	40.0	14.3–112.1	<0.001
**Variables**		**Multivariable Analysis (Ki-67% index)**		
		** *HR* **	** *95%CI* **	** *p-Value* **
Age at diagnosis		1.04	1.0–1.1	<0.001
Radical surgery (ref: yes)		3.8	2.0–7.0	<0.001
	Stage			0.012
	Local	1.0		
	Local and mesenteric lymph nodes	0.9	0.3–2.8	0.92
	Intra-abdominal metastases	2.2	0.7–6.4	0.16
	Extra-abdominal metastases	2.6	0.8–7.9	0.10
Ki-67%		1.2	1.1–1.4	<0.001

**Table 3 cancers-16-00204-t003:** Prognostic factors for recurrence after radical surgery (group2). Uni- and multivariable analysis of prognostic factors for recurrence-free survival (RFS) after radical surgery. Only variables with *p*-value in univariate analysis < 0.15 (except age) are included in the table and in the multivariable analyses. Hazard ratio (HR) for plasma chromogranin A (p-CgA) and Ki-67 index were estimated per 2-fold increase in non-logarithmically transformed p-CgA and Ki-67 index. CI: confidence interval.

**Variables**		**Univariable Analysis**		
		** *HR* **	** *95%CI* **	** *p-Value* **
Age at diagnosis (n = 223)		1.0	1.0–1.0	0.72
Stage (n = 223)				<0.001
	Local (n = 42)	1.0		
	Local and mesenteric lymph nodes (n = 159)	2.8	1.2–6.5	0.019
	Intra-abdominal metastases (n = 20)	6.8	2.6–17.8	<0.001
	Extra-abdominal metastases (n = 2)	8.7	1.0–73.5	0.05
Ki-67% (n = 214)		1.2	1.0–1.5	0.08
p-CgA (n = 208)		1.3	1.2–1.5	<0.001
5-HIAA (plasma/urin) (normal level as ref.) (n = 123)		2.3	1.2–4.6	0.014
**Variables**		**Multivariable Analysis**		
		** *HR* **	** *95%CI* **	** *p-Value* **
Stage				0.016
	Local	1.0		
	Local and mesenteric lymph nodes	2.9	1.0–8.2	0.045
	Intra-abdominal metastases	5.9	1.8–19.2	0.003
	Extra-abdominal metastases	9.4	1.0–87.7	0.05
p-CgA		1.3	1.1–1.5	0.001
Ki-67%		1.4	1.1–1.9	0.01

**Table 4 cancers-16-00204-t004:** Prognostic factors for disease-specific survival after radical surgery (group2). Uni- and multivariable analysis of prognostic factors for disease-specific survival after intended radical surgery. Only variables with *p*-value in univariate analysis < 0.15 (except age) are included in the table and in the multivariable analyses. Hazard ratio (HR) for plasma chromogranin A (p-CgA) and Ki-67 index were estimated per 2-fold increase in non-logarithmically transformed p-CgA and Ki-67 index. CI: confidence interval.

**Variable**		**Univariable Analysis**		
		** *HR* **	** *95%CI* **	** *p-Value* **
Age at diagnosis (n = 224)		1.0	1.0–1.1	0.16
Year of diagnosis, 2000–2010 vs. 2010–2021 (n = 224)		0.45	0.17–1.24	0.124
Stage (n = 224)				0.111
	Local (n = 43)	1.0		
	Local and mesenteric lymph nodes (n = 159)	1.3	0.4–4.5	0.70
	Intra-abdominal metastases (n = 20)	1.8	0.4–9.2	0.46
	Extra-abdominal metastases (n = 2)	15.4	1.5–157.2	0.021
p-CgA (n = 208)		1.3	1.0–1.6	0.045
**Variable**		**Multivariable Analysis**		
		** *HR* **	** *95%CI* **	** *p-Value* **
Stage				0.022
	Local	1.0		
	Local and mesenteric lymph nodes	2.9	0.4–22.6	0.32
	Intra-abdominal metastases	2.1	0.2–26.1	0.58
	Extra-abdominal metastases	67.1	3.7–1213.0	0.004
p-CgA		1.3	1.0–1.7	0.05

**Table 5 cancers-16-00204-t005:** Prognostic factors for disease specific survival in patients with unresectable disease (group3). Uni- and multivariable analysis of prognostic factors for disease-specific survival of patients with unresectable disease or remnant disease on first postoperative imaging. Only variables with *p*-value in univariate analysis < 0.15 (except age) were included in the table and in the multivariable analyses. Hazard ratio (HR) for plasma chromogranin A (p-CgA) and Ki-67 index were estimated per 2-fold increase in non-logarithmically transformed p-CgA and Ki-67 index. Since both WHO grade and Ki-67 index were significant risk factors in univariable analysis, two different multivariable analyses were conducted. CI: confidence interval.

**Variable**		**Univariable Analysis**		
		** *HR* **	** *95%CI* **	** *p-Value* **
**Age at diagnosis (n = 391)**		1.04	1.0–1.1	<0.001
**Primary tumor resection (ref: yes)**		2.0	1.5–2.7	<0.001
**Stage (n = 391)**				0.017
	Local (n = 4)	1.0		
	Local and mesenteric lymph nodes (n = 61)	1.2	0.2–9.2	0.86
	Intra-abdominal metastases (n = 251)	2.4	0.3–17.1	0.39
	Extra-abdominal metastases (n = 75)	3.2	0.4–23.7	0.25
**WHO Grading (n = 357)**				<0.001
	NET G1 (n = 100)	1.0		
	NET G2 (n = 238)	1.1	0.8–1.6	0.63
	NET G3 (n = 14)	1.4	0.5–3.5	0.52
	NEC (n = 5)	27.6	9.9–76.5	<0.001
**Ki-67% (n = 356)**		1.2	1.1–1.4	<0.001
**p-CgA (n = 360)**		1.2	1.1–1.3	<0.001
**5-HIAA (plasma/urine) (normal level as ref.) (n = 234)**		1.6	1.0–2.7	0.074
**Variable**		**Multivariable Analysis (WHO Grade)**		
		** *HR* **	** *95%CI* **	** *p-Value* **
**Age at diagnosis**		1.04	1.0–1.1	<0.001
**WHO Grading**				<0.001
	NET G1	1.0		
	NET G2	1.1	0.7–1.7	0.70
	NET G3	2.4	0.9–6.3	0.07
	NEC	166.1	43.0–641.7	<0.001
**p-CgA**		1.2	1.2–1.3	<0.001
**Variable**		**Multivariable Analysis (Ki-67% Index)**		
		** *HR* **	** *95%CI* **	** *p-Value* **
**Age at diagnosis**		1.04	1.0–1.1	<0.001
**Primary tumor resection**		1.5	1.0–2.3	0.041
**Ki-67%**		1.3	1.1–1.5	<0.001
**p-CgA**		1.2	1.1–1.3	<0.001

**Table 6 cancers-16-00204-t006:** Medical treatment entire cohort.

Medicinal Treatment	Total
	n (%)
Somatostatin analog (SSA)	381 (62)
Peptide Receptor Radionucleotide Therapy (PRRT)	140 (23)
IntronA	149 (24)
Streptozotocin + 5-flourouracil	64 (10)
Everolimus	28 (5)
Temozolomide	25 (4)
Capecitabine	12 (2)
Etoposide	15 (2)
Carboplatin	13 (2)
Other (atezolizumab, topotecan, irinotecan)	4 (1)

## Data Availability

Data are contained within the article.

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
