# Peer review of "Outlook for 615 Small Intestinal Neuroendocrine Tumor Patients: Recurrence Risk after Surgery and Disease-Specific Survival in Advanced Disease"

_cancers, 2024, doi:10.3390/cancers16010204_

Round 1
Reviewer 1 Report
Comments and Suggestions for Authors
Well conducted study.
G1 vs G2 could be solved searching for a different Ki67 cutoff just published in 2017 on Endocrine
Comments on the Quality of English LanguageWell conducted study.
G1 vs G2 could be solved searching for a different Ki67 cutoff just published in 2017 on Endocrine
Author Response
Ses attached file

Reviewer 2 Report
Comments and Suggestions for Authors
Dear colleagues.
It has been a pleasure to review the present research.
I have a couple of important concerns:
- Based on its long time study period, how have been considered the changes in pathological classifications along this time? I mean, WHO or AJCC classification in 2000 is different to current one, although it is possible that criterion for such diagnosis at different times were the same. Did you do a re-assessment of the pathologic reports using the current classification?
- Table 2 is ininteligible and it should be improved.
- Edition of figures should also be improved.
- Table 4 has to be also improved.
- No new risk factors have been investigated and the results in terms of risk factors were already known. I guess the bigger strength of the present research is its long time follow-up. I would make more emphasis in such information along the discussion section and try to make an update from other papers.
Comments on the Quality of English Language
Author Response
Ses attached file

Reviewer 3 Report
Comments and Suggestions for Authors
The manuscript offers crucial insights into small intestinal neuroendocrine tumors (siNET), challenging conventional expectations, notably with a 12-year median recurrence-free survival post-radical surgery. The study questions the WHO grading system, emphasizing Ki-67 as a pivotal prognostic factor. Despite acknowledged limitations, the findings gain strength from a robust, large cohort with long-term follow-up. The manuscript advances siNET knowledge, prompting discussions on increased incidence and practical long-term follow-up recommendations. I recommend accepting this manuscript, with attention to the following minor issues:
1. Consider expanding the discussion on the observed doubling of siNET incidence from 2000-2009 to 2010-2020. Are there any potential explanations for this increase, and how might it impact prognosis or treatment strategies?
2. The manuscript questions the value of the current WHO grading system for siNET, specifically the cutoff at 3% for Ki-67. It would be beneficial to discuss potential implications for clinical practice and future directions for refining grading criteria.
3. The manuscript suggests the importance of long-term follow-up for siNET patients. Could there be practical recommendations for the frequency and type of follow-up assessments based on the study findings?
Comments on the Quality of English LanguageThe language in the manuscript is generally clear and precise, effectively conveying complex medical information.
Author Response
Ses attached file

Round 2
Reviewer 2 Report
Comments and Suggestions for Authors